# MicroRNAs as Regulators of Cancer Cell Energy Metabolism

**DOI:** 10.3390/jpm12081329

**Published:** 2022-08-18

**Authors:** Natarajaseenivasan Suriya Muthukumaran, Prema Velusamy, Charles Solomon Akino Mercy, Dianne Langford, Kalimuthusamy Natarajaseenivasan, Santhanam Shanmughapriya

**Affiliations:** 1Department of Biotechnology, School of Biotechnology and Genetic Engineering, Bharathidasan University, Tiruchirappalli 620 024, Tamil Nadu, India; 2Heart and Vascular Institute, Department of Medicine, Department of Cellular and Molecular Physiology, Pennsylvania State University College of Medicine, Dauphin, PA 17033, USA; 3Medical Microbiology Laboratory, Department of Microbiology, Centre for Excellence in Life Sciences, Bharathidasan University, Tiruchirappalli 620 024, Tamil Nadu, India; 4Department of Neural Sciences, Lewis Katz School of Medicine, Temple University, Philadelphia, PA 19140, USA

**Keywords:** miRNA, cancer metabolism, TCA, glucose oxidation, pentose-phosphate pathway, fatty acid oxidation

## Abstract

To adapt to the tumor environment or to escape chemotherapy, cancer cells rapidly reprogram their metabolism. The hallmark biochemical phenotype of cancer cells is the shift in metabolic reprogramming towards aerobic glycolysis. It was thought that this metabolic shift to glycolysis alone was sufficient for cancer cells to meet their heightened energy and metabolic demands for proliferation and survival. Recent studies, however, show that cancer cells rely on glutamine, lipid, and mitochondrial metabolism for energy. Oncogenes and scavenging pathways control many of these metabolic changes, and several metabolic and tumorigenic pathways are post-transcriptionally regulated by microRNA (miRNAs). Genes that are directly or indirectly responsible for energy production in cells are either negatively or positively regulated by miRNAs. Therefore, some miRNAs play an oncogenic role by regulating the metabolic shift that occurs in cancer cells. Additionally, miRNAs can regulate mitochondrial calcium stores and energy metabolism, thus promoting cancer cell survival, cell growth, and metastasis. In the electron transport chain (ETC), miRNAs enhance the activity of apoptosis-inducing factor (AIF) and cytochrome c, and these apoptosome proteins are directed towards the ETC rather than to the apoptotic pathway. This review will highlight how miRNAs regulate the enzymes, signaling pathways, and transcription factors of cancer cell metabolism and mitochondrial calcium import/export pathways. The review will also focus on the metabolic reprogramming of cancer cells to promote survival, proliferation, growth, and metastasis with an emphasis on the therapeutic potential of miRNAs for cancer treatment.

## 1. Introduction

The uncontrolled growth of a tumor increases the demand for energy and metabolites in the cells, and both are achieved by metabolizing extracellular nutrients. The dysregulated uptake and catabolism of metabolites that supply energetically demanding biosynthetic fluxes is a hallmark of cancer metabolism [1], and increasing the understanding of how cancer cells meet these metabolic demands has been a consistent focus of research over the past several decades. Since the discovery of aerobic glycolysis in cancer cells by Warburg in the 1920s [2,3], the field of cancer metabolism has grown exponentially and has presented numerous therapeutic possibilities for fighting this disease. Glutamine metabolism provides the carbon and amino-nitrogen biomass that is required for protein, nucleotide, and lipid biosynthesis in tumor cells. In cancer cells, the uptake of exogenous lipids or lipoproteins and endogenous lipogenesis are upregulated to meet their demand for lipids. These metabolic changes are controlled by oncogenic signals including the myelocytomatosis oncogene cellular homolog (MYC), hypoxia-inducible factor 1-α (HIF-1α), protein kinase B (AKT), and 5′ AMP-activated protein kinase (AMPK). Additionally, scavenging pathways (e.g., autophagy) also maintain tumor cell growth by supplying metabolites to meet these requirements. Apart from these canonical regulations, microRNAs have recently emerged as key regulators of cancer metabolism.

MicroRNAs are small noncoding RNAs with an average length of 22 nucleotides. they are involved in the regulation of biological processes as posttranscriptional regulators of gene expression, RNA silencing, etc. [4]. To date, approximately 2200 conserved miRNAs have been identified and are reported to interact with mRNAs [5,6], implying the significance of miRNAs in essentially all developmental processes including cell growth, differentiation, metabolism, viral infection, and tumorigenesis [7]. Although they are highly significant for the normal functioning of the cells, miRNAs have also been investigated in depth in numerous pathological settings, with cancer certainly leading the way. These developments in miRNA profiling and sequencing techniques have overcome one of the most provocative challenges and led to the discovery that miRNA expression is dysregulated in human cancers and that its signatures could be used for understanding the miRNA classification and prognosis for cancer development [8]. As the chief regulators of the mRNAs, healthy miRNAs regulate the energy metabolism in tumors either directly or indirectly by regulating the genes and enzymes involved in metabolic pathways [9,10]. MicroRNAs have been reported to regulate a plethora of enzymes involved in glucose, fatty acid, and amino acid metabolism, which is prone to being reprogrammed in cancer cells to meet the heightened metabolic demands. Therefore, it is imperative to understand the role of miRNAs in these metabolic pathways to generate new therapeutic avenues to combat this disease and to identify potential biomarkers for cancer diagnosis. 

This review will discuss in detail the role of miRNAs in cancer with special emphasis on energy metabolism and the mechanistic role of mitochondrial calcium regulation in supporting the energy demands of cancer cells. We have also addressed the importance of metabolic reprogramming in cancer cells as a means of survival, proliferation, and metastasis and the regulatory role of miRNAs in cancer cell energy metabolism. This review will provide a better understanding of the miRNA-based regulation of cancer energy metabolism. 

## 2. Energy Metabolism in Cancer Cells

The energy metabolism differs significantly between cancer and normal cells. To meet their energy demands, healthy cells rely primarily on mitochondrial oxidative phosphorylation (OXPHOS), while cancer cells rely heavily on glucose and aerobic glycolysis. Although mitochondrial oxidative phosphorylation generates adenosine triphosphate (ATP) more efficiently than glycolysis, the cancer cells resort to this ineffective pathway for energy production in a phenomenon known as the Warburg effect [11,12,13]. Cancer cells can be distinguished from normal cells by their metabolic needs and metabolic regulation. Normal cells have in place a plethora of regulatory mechanisms like multiple feedback and feedforward regulatory loops and a capacity to undergo quiescence when deprived of nutrients, whereas the cancer cells’ characteristic of growth and proliferation results in nutrient uptake due to sustained bioenergetic demand [14]. As proposed by Warburg, there is a long-standing impression that cancer cells have enhanced glycolysis due to the impairment of mitochondrial OXPHOS, but this thought has been challenged by recent studies that show the mitochondrial OXPHOS to be intact in several cancer cells [14,15,16,17]. Furthermore, mitochondria are versatile organelles with an inert ability to produce ATP and can adjust their metabolic phenotype according to the energy requirement and macromolecular synthesis [18,19]. Therefore, it is conceivable that the cancer cell mitochondria indeed can switch between glycolysis and OXPHOS for their survival. Recent studies have proven that cancer cells devoid of mitochondrial DNA lack their tumorigenic potential [20]. It has also been found that the cells can regain this ability by acquiring healthy mtDNA from host stromal cells via the horizontal transfer of whole mitochondria [21] to recover respiratory function. There is scientific evidence that the glycolytic phenotype in cancer cells might be due to the overpowering of OXPHOS by active glycolysis rather than defects in the mitochondrial function itself [16,22,23]. There is also a hypothesis called the reverse Warburg effect that inhibits the function of OXPHOS in cancer cells. The so-called reverse Warburg effect states that epithelial cancer cells can induce aerobic glycolysis in carcinoma-associated fibroblasts (CAFs) to produce lactate, ketones, and pyruvate to replenish the TCA cycle in cancer cells for OXPHOS [24,25]. Recently, another theory was proposed wherein cancer cells deploy metabolic symbiosis that involves the transfer of glycolysis-derived lactate to oxygenated tumor environments as a means of fueling OXPHOS and avoiding the competition for glucose. The preferential utilization of lactate would indeed save the glucose for the cells in the hypoxic environment [26,27]. Several studies have provided compelling evidence that glutamine plays an important role in the cellular growth of many cancers [28,29]. Many cancer cells have been thought to undergo metabolic reprogramming to use glutamine for survival and proliferation [30,31]. Glutamine-dependent cancer cells take up glutamine as the preferred anaplerotic substrate and convert it into TCA cycle metabolites in a process called glutaminolysis [31]. Yet another recent study has postulated that cancer cells produce NADH in cytosol using carbohydrates, fatty acids, and glutamine and transport it to the mitochondria to fuel the ATP production using the mitochondrial electron transport complex (ETC) [32]. Thus, cancer cells can continuously reprogram their metabolism to adapt well to environmental pressures and alterations in their growth conditions. 

## 3. MicroRNAs Involved in Glucose Metabolism

### 3.1. MicroRNAs and Glucose Uptake

Glucose enters the cell through glucose transporters (GLUTs/SLC2A). The GLUT is the most important protein, controlling glycolysis, and 14 GLUTs have been identified so far [33]. GLUTs are found to be overexpressed in several tumors. Compared with the normal surrounding tissues, GLUT1, GLUT2, and GLUT3 are highly expressed in cancer tissues [34,35,36]. The potentially increased levels of GLUTs in malignant cells seem to facilitate accelerated glucose metabolism. Several factors can regulate GLUTs; for example, the ovarian hormone estrogen regulates GLUT by modulating its expression [37]. In addition, hypoxia and metabolic-stress-induced signaling pathways trigger GLUT expression [34,38]. Apart from these regulatory activities, GLUTs can be regulated by several miRNAs, and the GLUT-targeting miRNAs are thought to be downregulated to favor various cancer types including ovarian cancer, lung cancer, colon cancer, bladder cancer, prostate cancer, and renal cell carcinoma (Figure 1). For instance, miR-132 has been known to be a GLUT1 suppressor, and its expression was found to be downregulated in malignancy, which ultimately resulted in higher expression of GLUT1 and enhanced glycolysis [39]. Additionally, Lui et al. reported a decreased expression of miR-144 in lung cancer wherein it was found to increase the glucose uptake [40]. The overexpression of miR-138, miR-150, miR-199a-3p, and miR-532-5p has been connected with decreased GLUT1 levels. On the contrary, the downregulation of miR-19a, miR-19b, miR-130b, and miR-301a are linked to the enhanced expression of GLUT1 in renal cell carcinoma [41]. The 3′ UTR of GLUT1 is known to be directly targeted by miR-495, miR-340, miR-186, miR-22, miR-328, and miR-1291 [42,43,44,45,46]. In most cancers, the reporting miRNAs were found to be downregulated, resulting in the increased expression of GLUT1 and thus facilitating a metabolic switch in favor of tumor development [39,47,48]. GLUT3 is known to be directly regulated by miR195-5p by targeting the 3′-UTR [49]. The dysregulated metabolism in colorectal carcinoma was attributed to the altered expression of miR-19a and miR-133 and their role in regulating GLUT levels [50]. In pancreatic tumors, the upregulation of GLUT1 expression and increased glucose uptake was linked to the downregulation of miR-130b [51], and miR-129-5p was identified as directly targeting GLUT3 and acting as a suppressor of glucose metabolism and cell proliferation in gastric cancer (GC) cells [52]. It was also identified that the miR-129-5p/SLC2A3 axis exerts its suppressor function by regulating the PI3K-Akt and MAPK signaling pathways.

### 3.2. Regulation of Glycolysis by miRNAs

In addition to the GLUTs, other glycolytic enzymes are also regulated by miRNAs (Figure 1). Several studies have emphasized the miRNA-mediated regulation of glycolysis in tumor cells. The conversion of glucose to glucose-6-phosphate by the enzyme hexokinase (HK) marks the first step of glycolysis. Four different isoforms (HK1-HK4) of hexokinase have been identified so far [53,54,55]. HK1 and HK2 are high-affinity enzymes, and their distribution varies between different tissues. HK2 is overexpressed in a variety of tumors and is a characteristic feature of cancer cells [56,57,58,59]. The overexpression of HK2 is known to drive tumor cell proliferation by supporting aerobic glycolysis. Thus, HK2 is a critical regulator of the Warburg effect and is now emerging as an important target for cancer metabolism [60,61]. Primarily, HK2 was identified as being regulated by miRNAs by Fang et al.; interestingly they demonstrated HK2 to be regulated by miR-125a and miR-143 and to modify glucose metabolism and cell proliferation in lung cancer cells [62]. The regulation of HK2 by miRNA was further confirmed in head and neck squamous cell carcinoma (HNSCC)-derived cell lines [63]. In colon cancer cells, miR-143 was identified as targeting HK2 directly [64]. Apart from these studies, the role of miR-143 in modulating HK2 has been reported in various cancers like colon cancer, esophageal squamous cell carcinoma, lung cancer, cervical carcinoma, liposarcoma, bladder cancer, osteosarcoma, and gastric cancer [62,63,65,66]. A common subtype of myelodysplastic syndrome is 5q syndrome, which is characterized by the interstitial deletion of chromosome 5q. This locus is often deleted in several other malignancies and was also identified as the locus where the miR-143 gene resides [67,68]. These reports all illustrated the role of miR-143 in regulating HK2 and glucose metabolism in cancer cells.

HK2 expression is also known to be modulated by another miRNA, miR-199a-5p, which was identified to be downregulated in liver cancer cells by mechanism-mediated HIF1α overexpression. Decreased miR-199a-5p expression promoted glycolysis and lactate production via HK2 regulation [69]. In addition, HK2 has also been reported to be regulated by miR-155, which is found to be upregulated in various tumors. It also acts as an oncomiR and negatively regulates tumor-suppressive genes like TP53INP1, RhoA, and socs1, thus promoting malignant transformation and cancer progression [70,71]. Two distinct mechanisms have been proposed to describe how miR-155 upregulates the expression of HK2: (i) miR-155 enables the activation of STAT3, which in turn promotes the transcription of HK2; (ii) miR-155 represses miR-143 by targeting C/EBPβ (a transcriptional activator of mir-143) and subsequently enhances the expression of HK2. Both miR-143 repression and STAT3 activation are essential for miR-155 to enhance glycolysis in breast cancer cells [51]. This phenomenon has been observed in other cancers like liver and lung. Interestingly, miR-155 expression has been reported to be upregulated via JNK, nuclear factor-κB (NF-κB), and activator protein-1 (AP-1) pathways [72,73], implicating a relationship between inflammation and altered metabolism in cancer cells. The miRNA-dependent regulation of HK is not limited to HK2 because HK1 is targeted by miR-138 [63].

Other important intermediate steps in glycolysis are also known to be regulated by miRNAs. The enzyme glucose-6-phosphate isomerase (GPI) is known to be regulated by miR-200 and miR-302b/miR-17-5p in breast cancer cells [74] and chicken primordial cells [75], respectively. Phosphofructokinase 1 (PFK1), another glycolytic enzyme is also regulated by miRNA. For instance, miR135 was identified as targeting PFK1, inhibiting aerobic glycolysis, and suppressing tumor growth [76]. In NCI-H460 lung cancer cells, phosphofructokinase liver type (PFKL) was known to be regulated by the miR-128–PFKL–AKT axis. The regulation of PFKL resulted in a metabolic shift from glycolysis to oxidative phosphorylation (OXPHOS) in lung cancer cells [77]. The enzyme aldolase (Ald A) catalyzes the reversible aldol reaction in which fructose 1,6-bisphosphate is broken down into glyceraldehyde 3-phosphate and dihydroxyacetone phosphate. Ald A is known to be regulated by miR-122 in liver cells [78,79], the miR-15a/16-1 cluster in leukemia [80], miR-31, and miR-200a in Y79 retinoblastoma cells [81]. Glyceraldehyde 3-phosphate dehydrogenase (GAPDH) catalyzes the glycolytic step that releases reducing equivalent NADH. GAPDH is also regulated by miRNAs including miR-644a and miR-155 [82,83] in cancer cells. Cancer cells re-express the embryonic isoform of pyruvate kinase (PK), PKM2, which dephosphorylates phosphoenol pyruvate (PEP) to pyruvate. PKM2 is known to be upregulated in many tumors due to the downregulation of various miRNAs that downregulate PKM2. The microRNAs miR-133a, miR-133b, miR326, and miR-122 are known to directly regulate PKM2 [84,85,86]. In glioblastoma cells, the upregulation of PKM2 is directly correlated with the decreased expression of miR-326 [84]. Likewise, miR-122 targets PKM2 and inhibits HCC proliferation. It was also shown that the increased methylation of a miR-122 promoter in HCC attenuates its expression and relieves PKM2 suppression [87]. Additionally, the decreased expression of miR-133a and miR133b in tongue SCC was associated with the increased expression of PKM2 [86]. In colorectal cancer, miR-124, miR137, and miR-340 regulate the switch of PKM gene expression from PKM2 to PKM1 [88]. Under hypoxic conditions, miR-210 represses ISCU1/2, thus decreasing the activity of proteins involved in mitochondrial metabolism [89]. Hence, miR-210 represses mitochondrial respiration and might indirectly facilitate aerobic glycolysis. In prostate cancer PC3 cells, miR-124 has been reported to regulate the PKM2 gene and thus suppress cancer cell proliferation [90]. Other miRNAs, miR-99a, miR-124, miR-137, and miR-340, are known to indirectly regulate PKM2 [88,91]. Taking these findings together, it is very evident that the deviances in glucose metabolism play a crucial role not only in cancer survival but also in tumor metastasis.

### 3.3. MicroRNAs Involved in Lactate Metabolism

Another important step in glycolysis is the final step: catalysis by lactate dehydrogenase (LDH) which converts pyruvate to lactate. This step is crucial in deciding the fate of glucose, where pyruvate can either enter the TCA cycle to yield 36 ATP molecules or be converted to lactate to produce just 2 ATPs. High levels of lactate production are often associated with enhanced tumorigenesis, and therefore the regulation of LDH is critical in cancer cells [92,93,94]. In recent years, the miRNA-based regulation of LDH has been established in cancer cells (Figure 1). In the maxillary sinus and esophageal anaplasias, the subunit B of LDH (LDHB) was identified as being regulated by miR-375 [95]. In addition, miR-34a, miR-34c, miR-369–3p, miR-374a, miR-4524a/b, miR-323a-3p, miR-200c, miR-30d-5p, and miR-30a-5p regulate subunit A of LDH (LDHA) in breast, colorectal, and pancreatic cancer cell lines and osteosarcoma tissues [96,97,98,99,100,101]. In some circumstances, a point mutation was found to occur in the binding site of miR-374a in 3′UTR of LDHA. As a result of this point mutation, the miR-374a fails to suppress LDHA expression [99] and thus to show its inhibitory effects, leading to tumor development. Therefore, microRNAs miR-142-3p, miR-200c, miR-30a-5p, miR-33b, miR-323a-3p, miR-489 and miR-383 can be exploited as therapeutic possibilities to combat respective cancers [97,98,102,103,104,105].

Monocarboxylate transporters (MCTs) are membrane proteins that maintain the lactate, pyruvate, and ketone bodies fluxes in cells. There are four MCT isoforms (MCT1-MCT4) described in humans, and the isoforms differ in their cellular distribution. Similar to LDH, MCTs are also regulated by miRNAs (Figure 1). MCT1 is targeted by miR-29a, miR-29b, miR124, and miR-495 in pancreatic β cells [106,107]. MCT1 is regulated by miR-343-3p and promotes alterations in lactate and glucose flows. In addition, miR-342-3p overexpression significantly decreased cell proliferation, viability, and migration in breast cancer cell lines [108]. MCT4 is regulated by miR-145, which causes the accumulation of lactate within hepatocellular carcinoma cells (HCC) [109]. During the progression of malignant melanoma, the highly expressed protein basigin (Bsg) interacts with MCT1 and 4. Bsg is targeted by Let-7b and is known to inhibit the invasiveness of melanoma cells, potentially through the disruption of this interaction [110].

## 4. Amino Acid Metabolism and miRNAs

The metabolic networks of amino acids have widespread effects in cancer cells and are involved in protein biosynthesis and purine and pyrimidine synthesis; they also act as neurotransmitters and play a role in epigenetic modifications, ATP production, and the detoxification of ammonia by conversion to urea. Hence, any alterations in the amino acid metabolism play diverse roles in metabolic control and the regulation of the tumor microenvironment [111]. Apart from glucose, cancer cells exhibit increased glutamine intake and metabolism (glutaminolysis) (Figure 2). It is thought that the proliferative phenotype of the cancer cell is maintained by this adaptive accelerated glutamine metabolism as it provides substrates for lipogenesis and nucleic acid biosynthesis [1,112]. Glutamate and then α−ketoglutarate (α-KG) are formed from glutamine by glutaminase (GLS) and glutamate dehydrogenase (GDH), respectively. In addition, glutamine metabolism produces increased levels of succinate, fumarate, malate, NADH, and ATP.

The transcription factor c-Myc has been identified as repressing the expression of miR-23a and miR-24b, and that results in the increased expression of glutaminase and the upregulation of glutamine catabolism in cancer cells [113]. MicroRNA miR-203 was reported to regulate glutaminase protein and also to sensitize malignant melanoma cells to temozolomide chemotherapy [114]. Glutamate can also be converted directly to glutathione (GSH) by the enzyme glutamate-cysteine ligase (GCL). GCL is considered the rate-limiting enzyme in GSH synthesis and is known to be regulated by miR-18a and miR153 in liver cancer and glioblastoma [115,116]. Additionally, miR-450a is known to target a set of mitochondrial mRNAs, decrease glycolysis and glutaminolysis, and limit the metastatic potential of ovarian cancer cells [117].

Apart from glutaminolysis, the serine, glycine, and one-carbon (SGOC) metabolic network is said to drive cancer pathogenesis, and this metabolic network promotes the methylation of DNA/RNA and ATP synthesis in cancer cells [118,119]. Additionally, 3-phosphoglycerate is converted to serine and then to glycine by serine hydroxymethyltransferase 1 (SHMT-1) and SHMT-2 [120]. Meanwhile, miR-193b has been shown to target SHMT2 and reduce MCF-7 breast cancer cell growth. In contrast, the overexpression of miR-198 decreases the SHMT1 expression in lung adenocarcinoma and also inhibits cell proliferation, apoptosis, and cell cycle arrest [121]. In an esophageal squamous cell carcinoma xenograft mouse model, phosphoserine aminotransferase (PSAT1) was identified as being directly regulated by miR-340 [122].

The branched-chain amino acids (BCAAs) including leucine, isoleucine, and valine also play important roles in cancer cells. Branched-chain α-ketoacid dehydrogenase (BCKD) catalyzes the irreversible step in BCAA catabolism. In mammals, miR-29b prevents the translation of the dihrolipoyl branched-chain acyltransferase component of the BCKS complex [123]. Separately, miR-218 is known to negatively regulate branched-chain amino acid transaminase 1 (BCAT1) levels and increases the sensitivity of PC3 and DU145 PCa cells to cis-diaminedichoroplatinum treatment [124]. Further pieces of evidence suggest that miR-494 sensitizes colon cancer to 5-FU by targeting dihydropyrimidine dehydrogenase (DPYD), an enzyme involved in β-alanine metabolism [125]. In colorectal cancer, miR-29a, miR-21, and miR-30d are known to reduce sensitivity to 5-FU by targeting amino acid metabolism [125].

Apart from targeting enzymes of amino acid metabolism, miRNAs are known to target amino acid transporters. Amino acids are transported into a cell via specific or nonspecific transporters and antiporters. The membrane protein SLC1A5 strictly regulates the transport of glutamine into the cell. A wide variety of tumors overexpress SLC1A5. The exogenous expression of miR-137 and miR-122 markedly inhibited SLC1A5 expression in a dose-dependent manner and was identified as altering glutamine metabolism in cancer [126]. In Huh7 cells, the cationic amino acid transporter (CAT-1) was identified as being translationally repressed by miR-122 [127]. The glutamine transporters in plasma membrane ASCT2 are gatekeepers for several amino acids’ entry into the cytosol region. The expression of ASCT2 was found to be higher in cancer cells than in normal cells. In colorectal carcinoma, glioblastoma, neuroblastoma, and prostate and pancreatic cancers the expression of ASCT2 was elevated, and the miR-137 was relatively downregulated. It is conceivable that by downregulating the expression of ASCT2, miR-137 inhibits glutamine metabolism, which is critical for cancer cell proliferation and survival [126]. Collectively, these data show that miRNAs are capable of regulating multiple genes involved in amino acid metabolism, and it is therefore hard to ignore the fact that the miRNA regulation of amino acid metabolism can be explored to find new therapeutic opportunities in the treatment of various cancers. 

## 5. MicroRNAs Involved in the Regulation of Pentose Phosphate Pathway

The pentose phosphate pathway (PPP) is required for the synthesis of ribonucleotides and is a major source of NADPH. The PPP supplies NADPH and ribose 5-phosphate (R5P), the most important metabolites for cell survival and proliferation [128], thus playing a versatile role in cancer metabolic reprogramming (Figure 3). Various studies have talked about PPP directly or indirectly as promoting cell survival and proliferation [129,130]. The primary regulatory step in the PPP flux is the step catalyzed by glucose 6-phosphate dehydrogenase (G6PD). It involves the irreversible oxidation of G6P into 6-phosphogluconolactone with the generation of NADPH [131]. G6PD has been considered the pacesetter for NADPH production and is overexpressed in various cancers like hepatocellular carcinoma and breast and lung cancer [132,133,134,135,136]. The miR-122, a highly conserved liver-specific miRNA, negatively regulates G6PD. G6PD is a functional miR-122 target: the loss of miR-122 has been linked to an altered hepatic metabolic profile [137,138]. Another muscle-specific miRNA, miR-1, has also been shown to negatively regulate G6PD [139,140]. A combined reduction of miR-122 and miR-1 contributes to the dysregulation of glucose metabolism in hepatocellular carcinoma (HCC) and results in rapid tumor progression [141]. He et al. [101] show that in pituitary tumors, the overexpression of miR-1 suppressed cell growth by targeting G6PD and inhibited the cancer cell metabolism. In colorectal cancer cells, miR-124 directly targets phosphoribosyl pyrophosphate synthetase 1 (PRPS1) and ribose-5-phosphate isomerase-A (RPIA) and thus inhibits DNA synthesis and proliferation [142]. Another important enzyme in the PPP is 6-phosphogluconate dehydrogenase (6PGD). Studies have reported 6PGD upregulation in a wide variety of tumors [143,144]. In lung tumor cells, 6PGD has been characterized as a functional target of miR-206 and miR-613 [145], and miR-206 and miR-613 are known to regulate 6PGD expression and metabolic reprogramming in cisplatin-resistant ovarian and lung cancer cells [144]. Therefore, the differential regulation of PPP and how cancer cells circumvent the PPP regulation represent a novel target for the diagnosis and treatment of tumors.

## 6. Lipid Metabolism in Cancer and miRNAs

Lipids form a diverse group of non-water-soluble molecules and include triglycerides (TG), phospholipids (PL), sterols, and sphingolipids (SPL). TG serves as the major energy source, whereas PL, sterols, and SPL form the major structural components of biological membranes. Lipids are important signaling molecules; they function as second messengers and hormones. Increasing evidence in recent years shows cancer cells to specifically alter different aspects of lipid metabolism (Figure 4). One characteristic feature that marks cancer cells is that irrespective of the concentrations of extracellular lipids, *de novo lipogenesis* was identified as increasing [146]. Another characteristic is that some tumor types exhibit the increased oxidation of lipids as their energy source instead of higher rates of glycolysis. Prostate cancer generally has a lower rate of glucose metabolism with a concomitant increase in fatty acids uptake and oxidation [147,148]. Thus, the increased availability of lipids contributes to cell growth, proliferation, survival under oxidative and energy stress, chemoresistance, support of a high glycolytic rate by promoting the redox balance and stimulation of signaling pathways that lead to invasion and metastasis [145,149]. MicroRNAs including miR-122, miR-33, miR-27a/27b, miR-34a, miR-21, and miR-378 were shown to regulate lipid homeostasis in cancer cells [150,151,152,153,154,155]. In cancer cells, transcription factors like c-Myc, TGF-β, and NF-κB transcriptionally inhibit miR-29 [156,157,158]. Separately, miRNA-29 was identified as a regulator of the negative feedback mechanism that modulates SREBP-cleavage activating protein and sterol regulatory element-binding protein-1 (SCAP/SREBP-1) signaling in glioblastoma growth [159]. Treatment of glioblastoma xenografts with miR-29 significantly suppressed tumor growth by inhibiting the SCAP/SREBP-1 and lipogenesis [159]. PPAR co-activator 1-alpha (PGC-1α) plays a pivotal role in regulating cancer development [160]. PGC-1α is known to be inversely regulated by miR-217. In MCF-7 and MDA-MB-231, both mRNA and protein levels of PGC-1α are increased in correlation with a decrease in miR-217 expression [161]. Additionally, in a mouse model of esophageal squamous cell carcinoma xenograft, phosphoserine aminotransferase (PSAT1) was found to be directly regulated by miR-340 [115]. The miR-181a targets isocitrate dehydrogenase 1 (IDH1) and modulates the expression of genes involved in lipogenesis and β-oxidation [162]. Furthermore, miR-22 is known to inhibit ATP citrate lyase (ACLY) in osteosarcoma and cervical, prostate, and lung cancer cells [163]. Additionally, miR-33a/b has been reported as a master regulator of cholesterol/lipid metabolism. The miR-33a/b-encoding genes are present within the intron sequences of human *SREBF* genes, and miR-33a/b is known to regulate the expression of ATP-binding cassette transporter ABCA1 [164,165]. In this regard, SREBP and miR-33 co-operatively regulate cell proliferation and cell cycle progression [166]. Furthermore, miR33a/b also regulates lipid homeostasis by controlling the expression of genes involved in fatty acid oxidation and the energy homeostasis regulators like AMPK and SIRT6 [167]. It was shown that during hepatocellular transformation, the transient inhibition of HNF4α becomes a stable event, and the feedback loop consisting of miR-124, IL6R, STAT3, miR-24, and miR-629 maintains the transformed phenotype both in vitro and in vivo. Thus, the systemic administration of miR-124 is known to prevent and suppress HCC development in murine liver cancer model [168]. 

## 7. Mitochondrial Metabolism and miRNA

### 7.1. MicroRNAs Involved in the Regulation of the TCA Cycle

In tumor cells, aerobic glycolysis results in the conversion of glucose into pyruvate and subsequently into lactic acid. Acetyl CoA is committed to a truncated tricarboxylic acid (TCA) cycle, with the net outcome that acetyl CoA will be disseminated to the cytosol as citrate and cleaved by ATP citrate lyase (ACL) to produce oxaloacetate and acetyl CoA. Oxaloacetate is reduced to malate and imported back into the mitochondrial matrix and converted to oxaloacetate again in the matrix (generating NADH that serves to repress the TCA cycle). One of the common and widely accepted events in cancer cells is a shift in glucose metabolism from oxidative phosphorylation to aerobic glycolysis. Although cancer cells circumvent the TCA cycle and mainly rely on aerobic glycolysis for their energy demands, various studies endorse that cancer cells rely profoundly on the TCA cycle for their energy requirements. The TCA cycle can be described as the central metabolic hub of energy metabolism and the synthesis of macromolecules. The role of the TCA cycle and its regulation has been overlooked until recently [49,169] (Figure 4). Pyruvate dehydrogenase protein X component (PDHX) has been identified by Chen et al. as a direct target of miR-26a in colorectal cancer (CRC) cells, and miR-26a is known to modulate PDHX expression by directly targeting the 3′UTR of PDHX [170]. PDHX located in the mitochondrial matrix is a non-catalytic subunit of the PDH complex and is central to mitochondrial energy metabolism [171]. The PDH complex catalyzes the oxidative removal of glucose and pyruvate under aerobic conditions. Therefore, miR-26a has been shown to regulate glucose metabolism in colorectal cancer cells by inhibiting the conversion of pyruvate to acetyl CoA and entering the tricarboxylic acid cycle [170]. 

In addition, the enzyme pyruvate dehydrogenase B (PDHB) has been a known target of miR-370 and miR-146-5p in melanoma and colorectal cancer, respectively. The elevation of miR-370 and miR-146-5p has been reported to downregulate PDHB [172,173]. Other PDH subunit PDHA1 is regulated directly by Lin28A/Lin28B and indirectly by let-7. Let-7 is known to activate the PDH complex by directly inhibiting PDK1 [174]. Additionally, the suppression of isocitrate dehydrogenase 2 (IDH2) by miR-183 in glioma cells decreases the cellular levels of α-KG and in turn leads to an increase in aerobic glycolysis [175]. Several miRNAs, including miR-19a, miR-19b, miR-148a, miR-148b, miR-152, miR299-5p, miR-122a, miR-421, and miR-494, have been shown to regulate the citrate synthase gene. Another miRNA, miR-210, specifically induced by HIF-1α, represses the iron-sulfur cluster assembly proteins (ISCU1/2) [89]. It decreases the activity of the TCA cycle by facilitating the assembly of [4Fe-4S] and [2Fe-2S] iron-sulfur clusters, which are incorporated into the TCA cycle-related enzymes like aconitase. 

### 7.2. MicroRNAs in Mitochondrial Calcium Regulation in Cancer Cells

Although cancer cells are thought to exclusively rely on glycolysis for ATP production, they still require the oxidation of α-ketoglutarate (α-KG) in the mitochondria to produce the reducing equivalents essential for facilitating the reductive carboxylation pathway and and generating the metabolic intermediates [176]. Cancer cells tend to uptake higher amounts of the non-essential amino acid glutamine [177], which is then converted to glutamate by glutaminases and later to α-KG [178]. Then, α-KG enters the TCA cycle, where it serves as the substrate for α-KG dehydrogenase (α-KGDH). The activity of α-KGDH is strongly Ca^2+^ dependent [179]. Meanwhile, mCa^2+^ in its physiological state is known to activate three major mitochondrial matrix dehydrogenases, namely, pyruvate dehydrogenase (PDH), isocitrate dehydrogenase (ICDH), and α-KGDH. The regulation of these three key enzymes by Ca^2+^ has a strategic task in coordinating cellular workload and ATP generation [180,181,182]. Thus, cancer cells meet their energy demands by the activation of TCA cycle dehydrogenases through increased mCa^2+^ uptake [183]. On the contrary, pulmonary arterial cancer cells and colon cancer cells avoid mCa^2+^ overload to confer resistance to cell death. Oncogenes such as Ras and Akt are known to regulate apoptosis by modulating mCa^2+^ entry and thus inhibiting mCa^2+^ overload [184]. Although studies suggest mCa^2+^ uptake is a double edge sword and to regulate both metabolic shift and cell death pathways, mCa^2+^ signaling is considered to play a vital role in the progression of various cancers by promoting proliferation and cell migration, metastasis, and vascularization and conferring apoptosis resistance [183,185]. 

The outer mitochondrial membrane (OMM) has a large channel that is readily permeable to Ca^2+^. Ca^2+^ entry from the intermembrane space (IMS) to the mitochondrial matrix is primarily through a very selective, low-conductance Ca^2+^ channel known as the mitochondrial Ca^2+^ uniporter (MCU) [186,187]. The molecular identity of the channel was unknown for several years, and simultaneous publications from Rizzuto and Mootha’s laboratory revealed that *CCDC109A* is the pore-forming subunit of the MCU complex. That is the moment when the mCa^2+^ signaling field started to witness an abundance of discoveries aimed at elucidating both the composition and functionality of the MCU complex. Once known only as a phenomenon that could be inhibited by ruthenium red derivatives, the mCa^2+^ influx machinery has now grown into the multiprotein assembly known as the uniporter MCU complex. The Ca^2+^ that enters the mitochondria will be pumped back to the IMS by a Na^+^/Ca^2+^ exchanger (NCLX) [188,189,190,191,192,193,194] or the H^+^/Ca^2+^ exchanger (HCX). The role of HCX, otherwise known as LETM1 (the leucine zipper-EF-hand containing transmembrane protein 1), as an H^+^/Ca^2+^ exchanger or a K^+^/H^+^ exchanger is still debated. Thereby, mCa^2+^ is established by the dynamic equilibrium between MCU-mediated Ca^2+^ entry and NCLX/HCX-mediated Ca^2+^ extrusion. 

MicroRNAs are considered important regulators for mCa^2+^ uptake. In silico analysis has shown that five miRNAs, namely, miR-15, miR-17, miR-25, and miR-137, could target MCU and/or MICU1 in tumor cells [195]. Specifically, the inhibition of mCa^2+^ entry by cancer-related miR-25 represents the first evidence for the regulation of MCU by miRNA [195] and offers preliminary clues to the significance of miRNAs in regulating mCa^2+^ entry. For instance, miR-25 affects mCa^2+^ homeostasis by downregulating MCU, prompting a strong decrease in mCa^2+^ uptake and conferring resistance to Ca^2+^-dependent apoptotic challenges [195]. Prostate cancer cell lines expressing high levels of miR-25 display very low levels of MCU expression, and this same phenomenon (high miR-25 and low MCU) is also maintained in colon cancer cell lines. The inhibition of miR-25 in HCT116 cells increases mCa^2+^ levels and is known to re-sensitize the cells to apoptosis. Similarly, a connection has been established through the detection of high miR-25 and undetectable MCU levels in stage II and stage III colonic adenocarcinoma. Other members of the miR-25 family such as miR-92a and miR-363 have been found to have the same effect as miR-25 on the expression of MCU and Ca^2+^ signaling. These findings not only emphasize the profound involvement of the whole family of this miRNA in the regulation of mCa^2+^ homeostasis but also propose how MCU downregulation favors cancer cell survival. Furthermore, miR-25-5p that originates from the opposite arm of the same pre-miRNA and members of the same pre-miRNA cluster such as miR-106b were predicted to target MCU mRNA, although their activity has not yet been tested. In contrast to miR-25, the downregulation of miR-340 is correlated with increased MCU expression in breast cancer. By targeting MCU miR-340 increases the rate of glycolysis, resulting in a metabolic shift and promoting cell migration and invasion [196]. Pulmonary arterial hypertension (PAH) results in a cancer-like phenotype as it is characterized by increased pulmonary artery smooth muscle cell proliferation, migration, and apoptosis resistance [197]. In this model, the authors noted a decreased expression of MCU that resulted in increased cytosolic Ca^2+^ and diminished mCa^2+^. The decrease in MCU level was found to be associated with increased expression levels of miR-25 and miR-138. 

Although OMM is permeable to Ca^2+^, studies show VDAC to play a critical role in regulating mCa^2+^ uptake. In many cancer cells, the expression of VDAC1 is found to be profoundly high. The miR-7 that negatively regulates VDAC1 mRNA is downregulated in cervical cancer and hepatocellular carcinoma [198]. Both cervical cancer (CC) tissues and cell lines have shown higher expression of lncRNA SOX21 antisense RNA 1 (SOX21-AS1), a long noncoding RNA. This long noncoding RNA has been involved in CC cell proliferation, migration, and invasion. It was found that miR-7 interacts with the miRNA-binding site of SOX21-AS1 and that the overexpression of SOX21-AS1 decreases the expression of miR-7 in cervical cancer cells. That study also reported that miR-7 directly targets the 3′-UTR of VDAC1 [199]. Likewise, miR-320a was downregulated in NSCLC cells, and miR-490-3p was downregulated in colorectal cancer, both of which can directly target VDAC1 mRNA and regulate its expression [200,201]. In cervical cancer, miR-613 was downregulated, and it was found to directly target LETM1 in many cancer cells. Additionally, the expression of LETM1 was high [202]. 

### 7.3. MicroRNAs in Mitochondria-Mediated Apoptosis in Cancer Cells

Mitochondria-mediated apoptosis is completely regulated by intrinsic apoptotic signals. When cells undergo oncogenic stress, DNA damage, and uncontrolled proliferation, it triggers apoptosis. Thus, the inhibition of cell growth by apoptosis is an ideal cancer treatment [203,204]. The mitochondria-mediated apoptosis pathway involves the release of apoptosis-inducing factors (AIF) and cytochrome c from mitochondria, leading to caspase-dependent cell death. The AIF and cytochrome c act as a double-edged sword by contributing to the cell death pathway and also regulating mitochondrial energy metabolism by increasing the oxidative phosphorylation reaction (OXPHOS). Importantly the function of AIF includes the posttranscriptional regulation of complex 1 in the mitochondrial respiratory chain and of cytochrome c in transferring the electrons from complex 3 to complex 4 in the electron transport chain. In a wide variety of cancers, both mRNA and protein levels of AIF and cytochrome c are found to be elevated and to help in the survival of cancer cells with poor prognosis [205,206,207,208]. As previously discussed, in the mCa^2+^ metabolism of cancer cells, aerobic glycolysis is also an important contributor to cancer cell survival. To support this, the BAX and BAK present in OMM are suppressed by several miRNAs in cancer cells. BAX and BAK are the gateways for the release of AIF and cytochrome c from mitochondria into the cytosol of a cell [209]. Simultaneously, the anti-apoptotic proteins Bcl-2 and Bcl-x levels are elevated, and some miRNAs that directly target these anti-apoptotic proteins are suppressed. 

The expression of miR-365 was found to be high in pancreatic cancer cells; it directly targets the pro-apoptotic regulator BAX and thus prevents cancer cell death by blocking the release of AIF and cytochrome c [210]. Radiation-induced mouse thymic lymphomas show high expression of miR-467a and result in the downregulation of BAX [211]. The expression of BAK was significantly reduced in breast cancer, and elevated miR-125b was observed, which explains the inhibition of Taxol-induced cytotoxicity and apoptosis [212]. The downregulation of miR-574-3p and restoration of miR-574-3p induce apoptosis by reducing the anti-apoptotic protein Bcl-xl in prostate cancer cell lines and clinical PC tissues [213]. Similarly, the Bcl-xl targeting miRNAs like miR-608, miR-133a, miR-491-5p, let-7, and miR-491 have been reported to be downregulated in different cancers, causing aggressive cancer progression [214,215,216,217,218]. The expression of Bcl-2-targeting miRNAs was found to be decreased in MALT lymphomas and in diffuse large B-cell lymphoma (DLBCL) with higher expressions of Bcl-2 in stages 3 and 4 of both types of lymphomas. The decreased expression of miR-34a and increased FOXP1, p53, and BCL2 co-expression have been linked to poor overall survival in MALT lymphoma and DLBCL patients. The same outcomes were observed in different cancers, where miR-15, miR-16, miR-30b, miR-125a-5p, miR-182, and miR-206 were found to be downregulated. They directly target the 3’UTR sequence of Bcl-2 mRNA [219,220,221,222,223,224]. 

## 8. Conclusions

MicroRNAs have emerged as one of the key players in regulating cellular metabolic pathways. In the last 15 years, research in the field of miRNAs has brought us immense information on the roles of miRNA in cancer cell pathophysiology. It became evident from several studies that all described hallmarks of cancer are related to some miRNA imbalance [225]. In this review, we list a number of miRNAs that act on metabolic molecules and pathways and contributes to tumor metabolic reprogramming. Despite our improved understanding of the role of miRNAs on individual enzymes, proteins, and metabolic molecules, a detailed and deep understanding of the overall impact of miRNA-mediated metabolic effects on various hallmarks of tumor is still required. In most of the studies selected for this review, changes in specific miRNA were assessed by significant changes in metabolic reprogramming and in tumor size or volume (increase or decrease). The knowledge that these miRNAs regulate cancer metabolism was largely obtained on the basis of in vitro cell culture and mouse models. Althoug, ~60% of mouse miRNA loci are conserved in humans evolutionarily [226,227], the identification of a large proportion of “species-unique miRNAs” [228] questions the accuracy of the knowledge of miRNAs obtained on the basis of mouse models. Thus, the biological outcomes of these miRNAs need to be re-examined in species-dependent and global contexts. To translate these miRNAs to clinical trials, we also see the importance of non-rodent models, as the immunostimulation triggered by oligonucleotides (miRNAs) is significantly different in nature in rodents and primates. Though significant obstacles still lie in the way of using these miRNAs in clinical practice, the results from rodent studies are promising, and two experimental miRNA-based therapies are now listed on clinicaltrials.gov. Several other miRNAs are now being tested in clinical trials. Most of them are in phase I and II. Together with the efforts directed towards the generation of model systems, by exploring the facts of how miRNAs regulate mCa^2+^ homeostasis, mCa^2+^-mediated metabolic shift, and mCa^2+^-mediated cell death mechanism in cancer cells will further accelerate the identification of therapeutic agents that target mitochondria to efficiently and robustly treat cancer. 

## Figures and Tables

**Figure 1 jpm-12-01329-f001:**
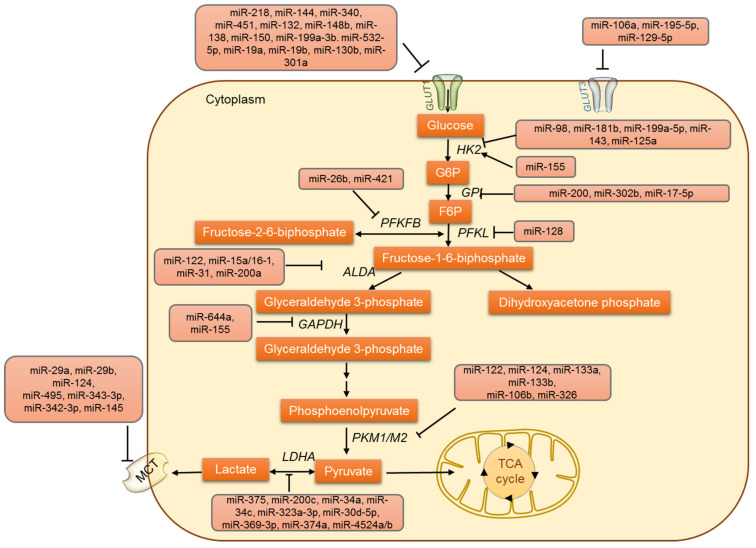
The regulation of glucose uptake, glucose oxidation and lactate metabolism by miRNAs.

**Figure 2 jpm-12-01329-f002:**
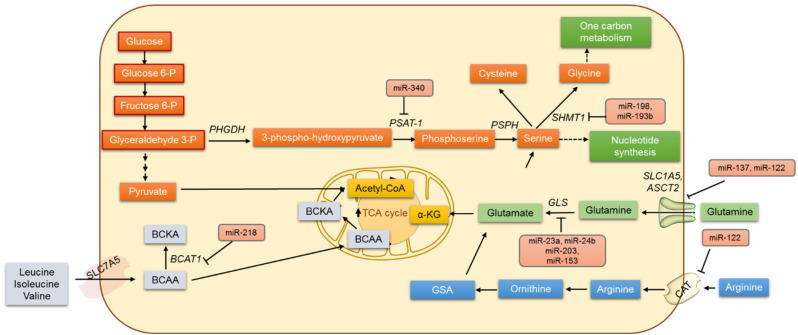
The regulation of amino acid cancer metabolism by miRNAs.

**Figure 3 jpm-12-01329-f003:**
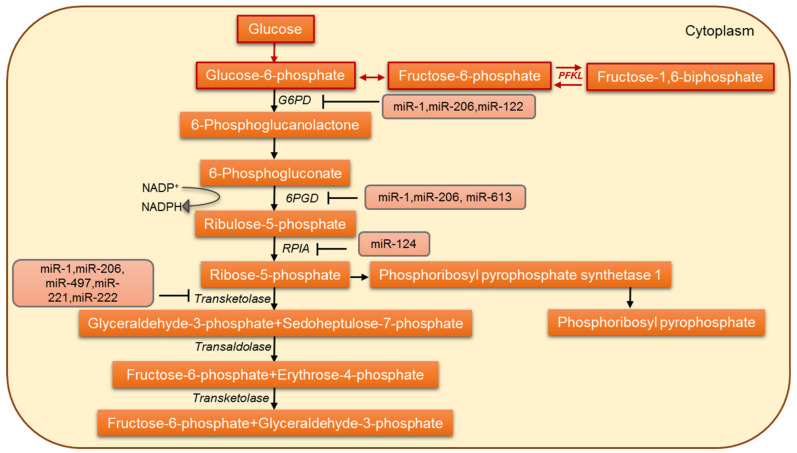
The miRNA-based regulation of the pentose-phosphate pathway in cancer cells.

**Figure 4 jpm-12-01329-f004:**
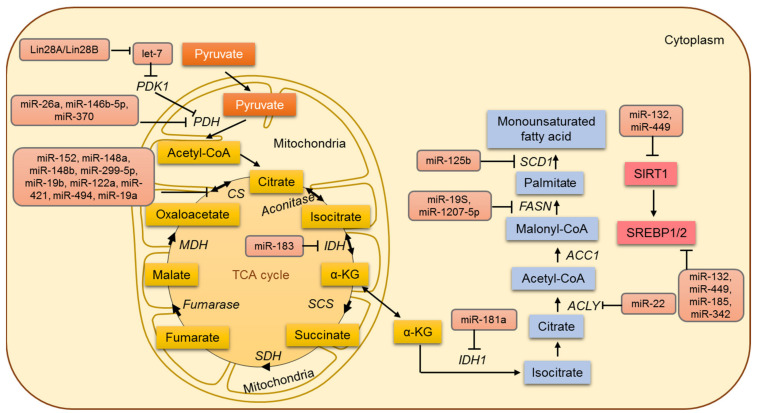
The miRNA-based regulation of lipid and mitochondrial metabolism in cancer cells.

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
