# Peer review of "MicroRNAs as Regulators of Cancer Cell Energy Metabolism"

_jpm, 2022, doi:10.3390/jpm12081329_

Round 1

Reviewer 1 Report

This is a comprehensive review describing the role of microRNAs in cancer cell energy metabolism regulation. My primary concern is that combining the result of very different organisms (human, mouse model, chicken...) and settings (healthy, cancer, other diseases) without any regard for the differences between them undermines the goal of the review. If the authors believe that ''Cancer is a heterogeneous disease and each cancer has its unique metabolic features.', how much sense does it make to discuss (and present in single figure) together studies of different cancer histologies, genetic backgrounds etc.? 

Other points:

1. The last paragraph of introduction needs to be re-written; to say that a review 'opens up numerous possibilities for further research' is a clear over-statement. 

2. Vague sentences that do not bring anything new should be deleted ('Cancer is a heterogeneous disease and each cancer has its unique metabolic features.' seems both vague and a strange introduction to a paragraph where authors discuss general metabolic features of all cancers; 'Several studies have provided compelling evidence that glutamine plays an important role in the cellular growth of many cancers.' - without any citation provided for any of the 'several studies'; 'The role of the TCA cycle and its regulation has been overlooked until recently')

3. 'Cancer cells can be distinguished from the normal cells with one metabolic difference where normal cells have in place a plethora of regulatory mechanisms like multiple feedback and feedforward regulatory loops and a capacity to undergo quiescence when deprived of nutrients' - what exactly is the single distinctive difference, 'a plethora of regulatory mechanisms', or 'a capacity to undergo quiescence'?

4. sentences beginning with 'but' or 'whereas' are confusing and should be merged with preceding ones (e.g. 'Whereas the cancer cells' characteristic of growth and proliferation results in nutrient uptake due to a sustained bioenergetic deman')

5. 'Whereas the cancer cells' characteristic of growth and proliferation results in nutrient uptake due to a sustained bioenergetic deman' - growth and proliferation in normal cells also results in nutrient uptake due to a sustained demand, so it does not seem like much of a difference between cancer and normal cells

6. strange phrases like 'scientific shreds of evidence', 'reporting' miRNAs (line 165) and use of tenses ('the enzyme pyruvate dehydrogenase B (PDHB) has been a known target of the miR-370 and miR-146-5p in melanoma and colorectal cancer respectively' - is it currently not a target of those miRNAs?) - I believe the review would benefit from a help of a native English speaker

7. The sentence 'Several factors can regulate GLUTs expression, for example, the ovarian hormone estrogen regulates GLUT by modulating its expression' is a tautology

8. 'In most cancers, the reporting miRNAs were found to be downregulated resulting in increased expression of GLUT1 and thus facilitating a metabolic switch in favor of tumor development.' (lines 165-167) - this is a very strong statement that should be backed up by citations accurately representing 'most cancers'. Are there any studies where they were found to be upregulated in any cancer?

9. How does miRNA regulation in chicken primordial cells (line 221) translate to human cancer? 

10. What is the quality of evidence on miRNA presence inside miRNAs? How are they transported via the mitochondrial membrane? These are crucial issues that should be discussed in detail before expanding on any miRNA role in regulating targets inside mitochondria.

11. I feel uncomfortable that no attention at all is given to the quality of evidence of the discussed studies, the validity and comparability of methods used. Without this, the review contributes very little.

Reviewer 2 Report

This review offers detailed information on the participation of various miRs in different aspects of metabolism in cancer. The topics are developed logically and following the approach of the introduction.

This work includes a lot of citations, but many of them are previous to the last five years, which may indicate that some miR studies of some miRs did not continue.

Schemes are well laid out and offer visual information about what is discussed in the text. The only suggestion I would like to make is that in cases where some information is being described that are not indicated in the diagrams, it is better to indicate it.

Or, on the contrary, indicate that information that is indicated in the figures, so that the reader wastes time looking for this information in the figures. Just as an example, lines 406-415 talk about the regulation of transporter ABCA1, AMPK, and SIRT6, but this information is not indicated in figure 4. It is just a small suggestion, for the reader; do not waste time trying to locate this information in the figure.

The information about MCU (lines 475-490) includes somewhat old quotes, I suggest reviewing and including more updated information:

doi: 10.1016/j.jbc.2022.101604.

doi: 10.1152/physrev.00041.2020.

doi: 10.1158/0008-5472.CAN-21-3230.

doi: 10.1016/j.isci.2021.102895

In general, I congratulate authors for the work done.

Reviewer 3 Report

In the present manuscript (jpm-1792282) Natarajaseenivasan Suriya Muthukumaran et al summarize and discuss the role of microRNAs in the regulation of cancer cell energy metabolism. This topic is relevant and of interest, and suitable for publication in the Journal of Personalized Medicine. Nonetheless the authors should consider the following minor comments:

Minor comments:

- Line 43: “In the electron transport chain” should be “In the electron transport chain (ETC)”

- Line 55: please consider “The uncontrolled growth of tumors increases the demand for energy and metabolites in the cells and both are achieved…” instead of “The uncontrolled growth of the tumor increases the demand for energy and metabolites in the cells and both of which are achieved…”

- Line 60: “Warburg in 1920” please include the reference.

- Line 73: please consider “biological processes as posttranscriptional” instead of “biological processes like posttranscriptional”.

- Line 81:”and led to the discovery of the fact that miRNA expression is unregulated in human cancers” Note: microRNAs have also been shown to be specifically down-regulated in cancer. Thus the sentence should be changed.

- Line 101: “Even in single cancer”. It does not sound well. Consider rewriting it.

- Line 108: “Cancer cells can be distinguished from the normal cells with one metabolic difference where normal cells have in place a plethora of regulatory mechanisms like multiple feedback and feedforward regulatory loops and a capacity to undergo quiescence when deprived of nutrients.” This sentence is confusing and should be rewritten.

- Line 157: “in malignancy”. Please specify the cancer type.

- Line 312: “carcinoma xenograft mouse model, phosphoserine aminotransferase” instead of “carcinoma xenograft, mouse model phosphoserine aminotransferase”.

- Line335: “higher” instead of “high”.

- Line 340: “these data show that miRNAs” instead of “these data show miRNAs”.

-Line 390:  please consider “shown” instead of “known”.

- Line 402: please consider “and modulates” instead of “and decreases and increases”.

- Line 450: please consider “shown” instead of “known”.

Line 575: please consider “to fight cancer” instead of “to fighting cancer as a disease”.

Round 2

Reviewer 1 Report

While most of my minor comments have been adressed, my main concerns remain. The authors state in their response that "miRNAs and the genes that they regulate are conserved in different organisms, so having an overview of the role of miRNAs in regulating cancer metabolism will help to design either miRNA based therapeutics or diagnostics that are common for most cancers.". This claim should be substantiated by any kind of evidence supporting evolutionary conservation for miRNAs that are reviewed based on animal studies. Evolutionary conservation between human and mice is known to not be perfect at least for some of the described miRNAs in a way that has important impact on the miRNA role in cancer (please see "Differential TGFβ pathway targeting by miR-122 in humans and mice affects liver cancer metastasis", Yin S. et al., Nature Communications 2016).

Regarding response to my last comment about the lack of discussion on the quality of evidence of reviewed studies and their comparibility, it is unaccaptable to simply state that addressing this issue is outside the focus of our study. I do not believe this review can be published without addressing this critical issue.

Author Response

We would like to thank the reviewers for their constructive comments and suggestions. Below we detail the changes that we have undertaken in the revised manuscript that address the reviewers’ queries.

While most of my minor comments have been adressed, my main concerns remain. The authors state in their response that "miRNAs and the genes that they regulate are conserved in different organisms, so having an overview of the role of miRNAs in regulating cancer metabolism will help to design either miRNA based therapeutics or diagnostics that are common for most cancers.". This claim should be substantiated by any kind of evidence supporting evolutionary conservation for miRNAs that are reviewed based on animal studies. Evolutionary conservation between human and mice is known to not be perfect at least for some of the described miRNAs in a way that has important impact on the miRNA role in cancer (please see "Differential TGFβ pathway targeting by miR-122 in humans and mice affects liver cancer metastasis", Yin S. et al., Nature Communications 2016).

Response: We thank the reviewer for this constructive comment. We appreciate the reviewer for highlighting this study. We agree with the reviewer that only 60% of the mouse miRNAs are conserved with human. Most of the studies and the miRNAs that are discussed in our review are taken from studies that use either in vitro cell culture or mouse model. Thus, it is highly recommended that the biological outcomes of these miRNAs need to be re-examined in species-dependent and global context. Though use of mouse model is a significant obstacle in cancer research, rodent studies are promising and two experimental miRNA-based therapies are now listed on clinicaltrials.gov. The drawback of the studies on which the review is based on and the need for non-rodent models to identify miRNAs that can translated to clinical trials are discussed in the conclusion section of the revised review article.  

Regarding response to my last comment about the lack of discussion on the quality of evidence of reviewed studies and their comparibility, it is unaccaptable to simply state that addressing this issue is outside the focus of our study. I do not believe this review can be published without addressing this critical issue.

Response: We agree with the reviewer. In most of the studies selected for this review, changes in specific miRNA was assessed by significant changes in metabolic reprogramming and in tumor size or volume (increase or decrease). The knowledge of these miRNAs regulating cancer metabolism was largely obtained on basis of in vitro cell culture or mouse models. When only ~60% of mouse miRNA loci are conserved in humans evolutionarily and identification of a large proportion of “species-unique miRNAs” questions the accuracy of the knowledge of miRNAs obtained on the basis of mouse models. Thus, the biological outcomes of these miRNAs need to be re-examined in species-dependent and global context. Despite our improved understanding on the role of miRNAs on individual enzymes, proteins, and metabolic molecules, a detailed and deep understanding of the overall impact of miRNA-mediated metabolic effects on various hallmarks of tumor is still required. The pitfalls of the studies carried out in identifying the miRNA in cancer metabolism are discussed in the conclusion section of the revised review article.